# LARGE LANGUAGE MODELS AS A COMPUTABLE SURROGATE TO SOLOMONOFF INDUCTION

## ABSTRACT

The rapid advancement of large language models (LLMs) calls for a rigorous theoretical framework to explain their empirical success. While significant progress has been made in understanding LLM behaviors, existing theoretical frameworks remain fragmented in explaining emergent phenomena through a unified mathematical lens. We establish the first formal connection between LLM architectures and Algorithmic Information Theory (AIT) by proving two fundamental results: (1) the **training process** computationally **approaches Solomonoff prior** through loss minimization interpreted as program length optimization, and (2) Under the assumption that $M(x_{1:t}) \approx \overline{M}(x_{1:t})$, LLMs' **next-token prediction** implements a form of **surrogate Solomonoff induction**. We leverage AIT to provide a heuristic, unified theoretical explanation for in-context learning, few-shot learning, and scaling laws. Furthermore, our theoretical insights lead to a principled method for few-shot example selection that prioritizes samples where models exhibit lower predictive confidence. We demonstrate through experiments on diverse text classification benchmarks that this strategy yields significant performance improvements, particularly for smaller model architectures, when compared to selecting high-confidence examples. Our framework bridges the gap between theoretical foundations and practical LLM behaviors, providing both explanatory power and actionable insights for future model development.

## 1 INTRODUCTION

Large Language Models (LLMs) have recently achieved significant advancements across multiple domains(Brown et al., 2020; OpenAI, 2024; DeepSeek-AI et al., 2025b; Qwen et al., 2025), and notably, their reasoning capabilities have improved substantially, as they can now generate intermediate reasoning steps, enhancing performance on complex tasks(Kojima et al., 2022; DeepSeek-AI et al., 2025a; Team, 2024; Team et al., 2025; He et al., 2025a). This unprecedented advancement has prompted researchers to seek theoretical frameworks that can systematically explain the emergent phenomena observed in these models(Wei et al., 2022; Nanda and Bloom, 2022; Wang et al., 2023; Meng et al., 2023; Delétang et al., 2024; Zheng et al., 2024; Ghandeharioun et al., 2024; Luo and Specia, 2024; Rai et al., 2025), yet providing a unified mathematical account for abilities like in-context learning(Dong et al., 2024), few-shot adaptation(Brown et al., 2020), and empirical scaling laws(Snell et al., 2024) remains a significant challenge for existing theories.

Foundational theories from computability and information theory offer potential avenues for deeper understanding. Notably, Algorithmic Information Theory (AIT)(Blum, 1967b;a) provides principles for universal sequence prediction based on algorithmic probability(Cover et al., 1989). Key concepts within this framework, such as the Solomonoff prior and Solomonoff induction—formalized concepts(Solomonoff, 1964a;b) originating from the work of Ray Solomonoff—offer a powerful lens for analyzing generative models. Their focus on sequence generation complexity provides a rigorous mathematical basis for universal prediction and inductive inference, thereby serving as the theoretical bedrock for our analysis.(Kolmogorov, 1965; Chaitin, 1966; 1977; Li et al., 2008; Downey and Hirschfeldt, 2010).

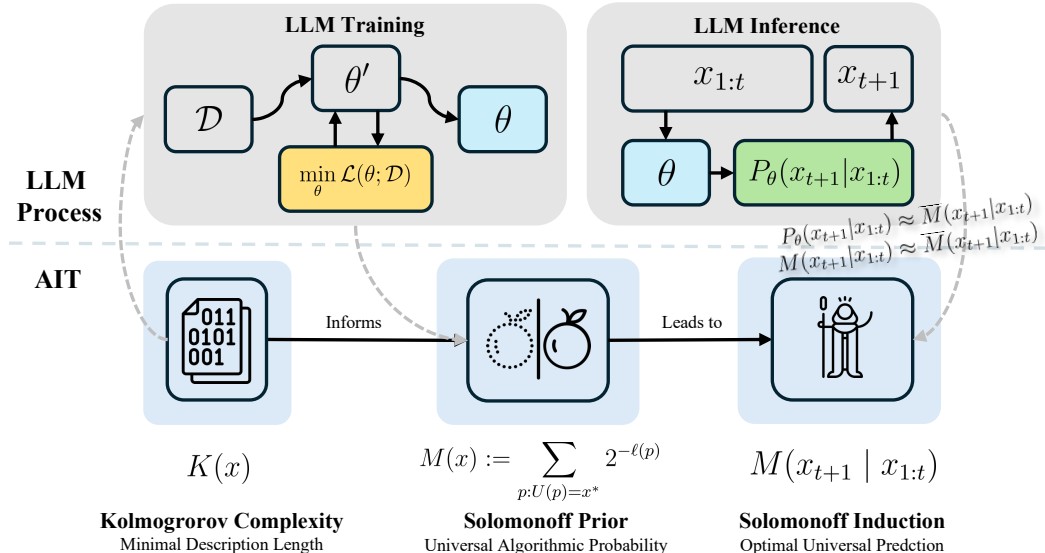

Figure 1: Conceptual diagram of our theoretical framework linking LLM processes to AIT. **LLM Process (Top):** Training optimizes parameters ($\theta'$) via loss minimization $\mathcal{L}(\theta; \mathcal{D})$, while inference uses $\theta$ to predict $x_{t+1}$ from $x_{1:t}$ via $P_\theta(x_{t+1}|x_{1:t})$. **AIT (Bottom):** Kolmogorov Complexity $K(x)$ informs the Solomonoff prior, which underlies Solomonoff induction $M(x_{t+1}|x_{1:t})$. Crucially, LLM training is shown to approaches the Solomonoff prior, and LLM inference's predictive distribution $P_\theta$ approximates Solomonoff induction (via $\overline{M}(x_{t+1}|x_{1:t})$) under the assumption that $M(x_{1:t}) \approx \overline{M}(x_{1:t})$, bridging LLM operations with AIT.

In this work, we forge a novel and rigorously established theoretical bridge between the operational principles of LLMs and the foundational concepts of AIT. We demonstrate that LLMs can be understood not merely as statistical pattern matchers but as practical, computable instantiations of Solomonoff's idealized framework for universal induction. Our primary contribution is a constructive mathematical proof that, within the standard LLM training paradigm, the minimization of prediction loss causes the model to asymptotically approach the Solomonoff prior, although full convergence remains unproven. This is achieved by reinterpreting the optimization process as a search for the shortest programs capable of generating the training data, thereby intrinsically linking learning efficiency to algorithmic compressibility.

Building upon this, we develop a formal argument showing that the next-token prediction mechanism inherent in LLMs forms a computable surrogate of Solomonoff induction, providing a theoretical underpinning for their generalization by casting predictive power as principled inductive inference; this connection yields a unified lens through which diverse emergent LLM behaviors can be understood as natural consequences of a system acting as a surrogate for universal induction. Guided by these AIT-based insights—particularly the inductive optimality properties of Solomonoff induction—we introduce and empirically validate a principled method for selecting few-shot demonstration examples: prioritize data points that expose the model's current predictive weaknesses (i.e., lower-confidence correct predictions) over those reinforcing well-learned patterns. Our experiments on SMS spam detection, emotion recognition, and news categorization with LLMs consistently show that prioritizing lower-confidence samples yields significant improvements over selecting high-confidence examples. Collectively, these contributions both advance a deeper theoretical understanding of LLMs and also offer actionable insights for their continued development and application.

## 2 RELATED WORKS

AIT builds on three foundational contributions: Solomonoff's universal prediction framework(Blum, 1967b;a; Cover et al., 1989), Kolmogorov's complexity metric(Kolmogorov, 1965), and Chaitin's incomputability results(Chaitin, 1966). Recent advancements in machine learning have explored the integration of Solomonoff induction into neural networks to enhance rapid learning from limited data.

(Grau-Moya et al., 2024) Similarly, This builds on established connections between deep learning generalization and AIT, where minimal description length models exhibit superior generalization(Blier and Ollivier, 2018). The compression perspective has become central to language modeling research, with studies demonstrating LLMs implicitly implement compression strategies(Everitt and Hutter, 2018; Lu et al., 2021; Delétang et al., 2024).

## 3 PRELIMINARIES

### 3.1 TURING MACHINES, NEURAL NETWORKS, AND LARGE LANGUAGE MODELS

The Turing machine (TM), introduced by Alan Turing in 1936 (Turing et al., 1936), is a foundational model of computation. Turing machines encompass both specific Turing machines, denoted $T$, and universal Turing machines (UTMs), denoted $U$. A UTM $U$ can simulate any other TM by processing a program $p$ and its input $w$ as arguments, denoted $U(p, w)$.

From a theoretical perspective, large language models (LLMs) can be viewed as specific Turing machines. Given an input context $x_{1:t}$, an LLM processes this sequence through a deep neural network to produce a conditional probability distribution $P(x_{t+1} \mid x_{1:t})$ over the vocabulary $\mathcal{V}$. A decoding strategy (e.g., greedy search, beam search, or temperature sampling) then generates the next token $x_{t+1}$ from this distribution. While the output appears stochastic due to sampling, the process is driven by deterministic pseudo-random number generators. Thus, the LLM as a whole functions as a deterministic Turing machine.

> **Definition**
>
> **Definition 1 (Language Model Generation Function)** *Let $\mathcal{X}$ be the set of input prompts, $\mathcal{S}$ the set of random seeds, and $\mathcal{R}$ the set of possible model outputs.*
>
> $$g : \mathcal{X} \times \mathcal{S} \to \mathcal{X}^* \tag{1}$$
>
> *such that for any $x \in \mathcal{X}$ and $s \in \mathcal{S}$, $g(x, s) = x^* = x \circ r$, where $r \in \mathcal{R}$ is the output generated by the model given $x$ and $s$, and $\circ$ denotes string concatenation.*

According to the above definition, $x^*$ is the full generated sequence starting with prompt $x$ and extended by $r$. The random seed $s$ influences both the semantic content and the length $|x^*|$ of the output.

### 3.2 PREFIX KOLMOGOROV COMPLEXITY

A prefix Universal Turing Machine (prefix UTM) is a fundamental construct in AIT. It processes input programs that form a prefix code—meaning no valid program is a prefix of another—ensuring unambiguous decoding of each input without requiring explicit delimiters.

The prefix Kolmogorov complexity (Li et al., 2008) quantifies the intrinsic information content of an object (e.g., a string or number). Formally, it is defined as the length of the shortest program that, when executed on a prefix UTM $U$, produces the object. For a prefix UTM $U$, the prefix Kolmogorov complexity of a string $x$ is defined as:

$$K_U(x) = \min\{\ell(p) : U(p) = x\} \tag{2}$$

where $p$ represents a binary program, $\ell(p)$ denotes its length in bits, and $U(p) = x$ indicates that $U$ halts and outputs $x$ when given input $p$. Conceptually, $K_U(x)$ represents the minimal descriptive complexity of $x$. Strings with inherent patterns or structure can be generated by concise programs, resulting in lower complexity values, whereas algorithmically random strings lack compact descriptions and exhibit higher complexity. Prefix Kolmogorov complexity exhibits two crucial properties: (1) While $K_U(x)$ depends on the specific choice of $U$, the difference between complexities measured under different prefix UTMs is bounded by a constant independent of $x$. Consequently, the subscript $U$ is frequently omitted in notation (*Invariance Theorem*). There exists no algorithm that can compute $K(x)$ precisely for all arbitrary strings, making it a non-recursive function (*Uncomputability*).

## 3.3 SOLOMONOFF PRIOR AND SOLOMONOFF INDUCTION

The Solomonoff prior (Solomonoff, 1960), introduced by Ray Solomonoff in the 1960s, is a foundational idea in AIT. It formalizes universal induction, a theoretically optimal method for inductive inference. The Solomonoff prior $M$ assigns a probability to any binary string $x$ as:

$$M(x) := \sum_{p:U(p)=x^*} 2^{-\ell(p)} \tag{3}$$

where $\ell(p)$ denotes the length (in bits) of program $p$, $x^*$ represents any string with prefix $x$, and $U$ is a prefix universal Turing machine (prefix UTM). The summation encompasses all programs $p$ such that $U(p)$ outputs a string beginning with $x$. This formulation embodies Occam's razor, as shorter programs contribute more significantly to $M(x)$. Two critical design choices warrant explanation: (1) The inclusion of outputs beginning with $x$ facilitates prediction—having observed sequence $x$, we aim to infer its continuation; (2) The prefix condition ensures that $M$ constitutes a semi-measure, satisfying $\sum_x M(x) \leq 1$, which is essential for probabilistic interpretation. Given an observed sequence $x_{1:t}$, Solomonoff induction defines the predictive probability for the next bit as:

$$M(x_{t+1} \mid x_{1:t}) = \frac{M(x_{1:t+1})}{M(x_{1:t})} \tag{4}$$

This framework has strong theoretical guarantees. Although the Solomonoff prior is uncomputable, it is semi-computable (Hutter, 2005), meaning we can approximate it increasingly well using computable functions.

## 4 MAIN RESULTS

### 4.1 THE TRAINING PROCESS OF LLMS AS A COMPUTABLE SURROGATE OF THE SOLOMONOFF PRIOR

> **Theorem**
>
> **Theorem 2 (LLM Training Approaches Solomonoff Prior)** *Let $\overline{f}(x,s)$ be a program constructed according to Definition 1, and define the approximate Solomonoff prior*
>
> $$\overline{M}(x) := \sum_{s=1}^{\infty} 2^{-\ell(\overline{f}(x,s))}$$
>
> *where $\ell(\overline{f}(x,s))$ denotes the length of the program describing $\overline{f}(x,s)$. Then:*
>
> 1. ***Upper Bound:*** $\overline{M}(x) \leq M(x)$, *where $M(x)$ is the Solomonoff prior.*
> 2. ***Monotonic Approach:*** *As the loss of $f$ decreases, $\overline{M}(x)$ increasingly approaches $M(x)$.*

As discussed in Section 3.1, large language models (LLMs) can be viewed as specific instances of Turing machines. Consequently, training an LLM can be interpreted as the process of identifying a Turing machine that best explains the observed data. In this section, we present a constructive argument demonstrating that the training process of LLMs is mathematically equivalent to a computable surrogate of the Solomonoff prior.

For any given string $x$, we can construct a program $f$ such that a universal Turing machine $U$ satisfies $U(f) = x$. This program $f$ comprises several components. The core model component includes the weight parameters of the LLM, inference logic, and sampling algorithm, with its binary representation denoted as $m_{(2)}$. Based on the theoretical work on language modeling as compression(Delétang et al., 2023), the compression and encoding component uses the LLM in conjunction with arithmetic coding to losslessly compress the string $x$, resulting in a binary encoding $e(x)_{(2)}$. Additionally, the decoding control component specifies the number of iterations $n(x)$ needed to decode $e(x)_{(2)}$ back to the original string $x$ with its binary representation denoted as $n(x)_{(2)}$. Finally, the random generation component provides a random seed $s$, with binary representation $s_{(2)}$, required by the LLM to generate subsequent content based on $e(x)_{(2)}$.

⭐ **Takeaway 1**: The LLM training process, driven by loss minimization, can be interpreted as an implicit search for programs of minimal algorithmic complexity that generate the training data, directly linking learning efficiency to data compressibility.

In summary, given a string $x$ and random seed $s$, the program $f$ can be represented as a 4-tuple:

$$f(x,s) = (m_{(2)}, n(x)_{(2)}, s_{(2)}, e(x)_{(2)}) \quad (5)$$

The execution of this program on the universal Turing machine $U$ proceeds as follows: (1) Based on model parameters $m_{(2)}$ and compressed code $e(x)_{(2)}$, the machine performs $n(x)$ iterations to restore the original string $x$ (*decoding phase*); (2) using $m(x)_{(2)}$, the restored $x$, and random seed $s_{(2)}$, the machine samples to generate the continuation of the output sequence $x^* \setminus x$ (*generation phase*). By combining the decoded $x$ with the generated continuation, the final output is the complete sequence $x^*$. This construction $f(x,s)$ has the following two key properties: (1) For a fixed input $x$, the number of decoding iterations $n(x)$ is deterministic; the random seed $s$ can be any natural number. (2) Since the model parameters $m_{(2)}$ remain fixed after training, they can, by Lemma 4, be internalized into the universal Turing machine $U$. As a result, the program can be simplified to $f(x,s) = (n(x)_{(2)}, s_{(2)}, e(x)_{(2)})$.

Since the Solomonoff prior is defined over a prefix universal Turing machine, we must encode $n(x)_{(2)}$, $s_{(2)}$, and $e(x)_{(2)}$ as prefix codes. To achieve this, we apply **Elias gamma coding** to each component, yielding the prefix-encoded representation:

$$\overline{f}(x,s) = (\overline{n}(x)_{(2)}, \overline{s}_{(2)}, \overline{e}(x)_{(2)}) \quad (6)$$

It can be shown that $\overline{f}(x,s)$ constitutes a valid set of prefix codes.

Let $\overline{F}$ denote the set of all such prefix-encoded programs. Based on this construction and Lemma 5, we define a prefix universal Turing machine $U_F$ with the following properties:

- $U_F$ is a prefix universal Turing machine;
- For any $\overline{f} \in \overline{F}$, we have $U_F(\overline{f}) = U(\overline{f})$.

Hence, for all $x$ and $s$, the following holds:

$$U_F(\overline{f}(x,s)) = x^* \quad (7)$$

Next, we define the **computable prior** $\overline{M}(x)$ as:

$$\overline{M}(x) := \sum_{s=1}^{\infty} 2^{-\ell(\overline{f}(x,s))} \quad (8)$$

Clearly, the quantity $\overline{M}(x)$ forms a subset of the Solomonoff prior $M(x)$, implying that $\overline{M}(x) \leq M(x)$. For any given $x$, since $n(x)_{(2)}$ is fixed, minimizing the binary encoding length $|e(x)_{(2)}|$ is crucial for making $\overline{M}(x)$ as close as possible to $M(x)$. As established in Delétang et al. (2023); Wan (2025), this objective aligns with minimizing the training loss of the large language model (LLM). Consequently, the training of an LLM can be viewed as a monotonic, computable approach to the Solomonoff prior $M(x)$, though we cannot prove that the convergence is arbitrarily precise.

### 4.2 THE INFERENCE PROCESS OF LLMs AS A COMPUTABLE SURROGATE OF SOLOMONOFF INDUCTION

As discussed in Section 4.1, $\overline{M}(x)$ can be viewed as a computable surrogate for the Solomonoff prior $M(x)$, although we cannot prove that it converges arbitrarily closely to $M(x)$. Under the assumption that $M(x_{1:t}) \approx \overline{M}(x_{1:t})$, we derive the following equation:

$$M(x_{t+1} \mid x_{1:t}) = \frac{M(x_{1:t+1})}{M(x_{1:t})} \approx \frac{\overline{M}(x_{1:t+1})}{\overline{M}(x_{1:t})} := \overline{M}(x_{t+1} \mid x_{1:t}) \quad (9)$$

Given the properties of Elias gamma coding, the code length for encoding a natural number $n$ is $\lfloor \log_2 n \rfloor + 1 + \lfloor \log_2 n \rfloor \approx 2 \log_2 n$ bits. Leveraging this property, we can express the code lengths

for Equation 6 as follows:

$$\left|\overline{s}_{(2)}\right| \approx 2\log_2 s$$

$$\left|\overline{n}(x)_{(2)}\right| \approx 2\log_2 n(x) \tag{10}$$

$$\left|\overline{e}(x)_{(2)}\right| \approx \left|e(x)_{(2)}\right| + 2\log_2\left|e(x)_{(2)}\right|$$

These relationships enable us to derive the following theorem.

> **Theorem**
>
> **Theorem 3 (LLM Inference Approximates Solomonoff Induction)** *Under the assumption that $M(x_{1:t}) \approx \overline{M}(x_{1:t})$, the conditional next-token probability $P_\theta(x_{t+1}|x_{1:t})$ approximates the Solomonoff inductive inference $M(x_{t+1}|x_{1:t})$. Asymptotically for large context length t, the relationship is given by:*
>
> $$M(x_{t+1}|x_{1:t}) \approx \overline{M}(x_{t+1}|x_{1:t}) \approx \frac{t^2}{4(t+1)^2} \cdot P_\theta(x_{t+1}|x_{1:t}) \tag{11}$$

*Proof.* We begin by computing the prior probability of the sequence $x_{1:t}$ using Equation 6 and Equation 10:

$$\overline{M}(x_{1:t}) = \sum_{s=1}^{\infty} 2^{-l(\overline{f}(x_{1:t}, s))} \tag{12}$$

$$\approx \sum_{s=1}^{\infty} \frac{1}{s^2} \frac{1}{t^2} \frac{1}{\left|e(x_{1:t})_{(2)}\right|^2} 2^{-\left|e(x_{1:t})_{(2)}\right|} \tag{13}$$

$$= \frac{\pi^2}{6} \frac{1}{t^2} \frac{1}{\left|e(x_{1:t})_{(2)}\right|^2} 2^{-\left|e(x_{1:t})_{(2)}\right|} \tag{14}$$

Similarly, for the sequence $x_{1:t+1}$:

$$\overline{M}(x_{1:t+1}) = \sum_{s=1}^{\infty} 2^{-l(\overline{f}(x_{1:t+1}, s))} \tag{15}$$

$$\approx \frac{\pi^2}{6} \frac{1}{(t+1)^2} \frac{1}{\left|e(x_{1:t+1})_{(2)}\right|^2} 2^{-\left|e(x_{1:t+1})_{(2)}\right|} \tag{16}$$

Thus, based on the Equation 9, we have:

$$\overline{M}(x_{t+1} \mid x_{1:t}) \approx \frac{t^2}{(t+1)^2} \frac{\left|e(x_{1:t})_{(2)}\right|^2}{\left|e(x_{1:t+1})_{(2)}\right|^2} \frac{2^{\left|e(x_{1:t})_{(2)}\right|}}{2^{\left|e(x_{1:t+1})_{(2)}\right|}} \tag{17}$$

On one hand, when $t$ is large, $\frac{\left|e(x_{1:t})_{(2)}\right|^2}{\left|e(x_{1:t+1})_{(2)}\right|^2} \approx 1$ On the other hand, $\left|e(x_{1:t})_{(2)}\right| \approx 2t - \sum_{i=1}^{t} \log_2 P(x_i \mid x_{1:i-1})$, where $P(x_i \mid x_{1:i-1})$ is the LLMs' predicted probability for the next token.

Combining the analysis above, we arrive at the key surrogate:

$$\overline{M}(x_{t+1} \mid x_{1:t}) \approx \frac{t^2}{4(t+1)^2} P(x_{t+1} \mid x_{1:t}) \tag{18}$$

$\square$

It should be noted that $M(x)$ is a semi-measure and needs to be normalized when converting to a probability. Since $\frac{t^2}{4(t+1)^2}$ is a value independent of the token, it will automatically cancel out during normalization. This result implies that, under the assumption $M(x_{1:t}) \approx \overline{M}(x_{1:t})$, the next-token probability predicted by an LLM can be viewed as a computable surrogate to Solomonoff's inductive inference.

### 4.3 Explaining Various Phenomena in LLMs using Solomonoff prior

Let $\mu$ be a computable target probability distribution. We have the following theorem (Hutter, 2005):

$$\sum_{t=1}^{\infty} \sum_{x_{1:t} \in \mathbb{B}^t} \mu(x_{1:t}) \Big( M(0 \mid x_{1:t}) - \mu(0 \mid x_{1:t}) \Big)^2 \leq \frac{1}{2} \ln 2 \cdot K(\mu) + c < \infty \tag{19}$$

Here, $\mu$ denotes the target distribution, $M(0 \mid x_{1:t})$ is the predictive probability of the next bit being 0 based on Solomonoff induction, and $\mu(0 \mid x_{1:t})$ is the predictive probability under the target distribution given the same conditions. $K(\mu)$ denotes the prefix Kolmogorov complexity of $\mu$. Since $\mu$ is computable, the term $\frac{1}{2} \ln 2 \cdot K(\mu) + c$ is finite. $\mathbb{B}^t$ denotes the set of all binary strings of length $t$.

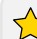 **Takeaway 2**: An LLM's next-token prediction is not merely statistical pattern matching but an surrogate of optimal inductive inference, offering a theoretical basis for its remarkable generalization capabilities on unseen sequences.

For the above infinite series to converge, its terms must approach zero. Specifically, this means that as $t \to \infty$, the prediction error $M(0 \mid x_{1:t}) - \mu(0 \mid x_{1:t})$ almost surely (with probability 1 under $\mu$) converges to zero. Therefore, the Solomonoff prior $M$ will eventually converge to the target probability distribution $\mu$.

We have previously noted that large language models (LLMs) can be viewed as computable surrogate of the Solomonoff prior $M$. Thus, leveraging the above theorem, we can attempt to explain several important phenomena observed in large language models:

1. In-context learning phenomenon: Since $M$ is a universal prior, for any computable target distribution $\mu$, it is possible to carefully design a context $x_{1:t}$ such that $M(0 \mid x_{1:t})$ approximates $\mu(0 \mid x_{1:t})$, thereby achieving learning effects.

2. Few-shot learning phenomenon: Adding a few examples (few-shot examples) in the prompt can sometimes significantly improve model performance. These examples increase the value of $\mu(x_{1:t})$, thereby giving them a higher weight in the error term of the theorem and accelerating the convergence of $M(0 \mid x_{1:t})$ to the target distribution.

3. Parameter scaling laws: Improving model performance by increasing the number of parameters is essentially a more precise surrogate of the Solomonoff prior through higher expressive capacity.

4. Inference scaling laws: Enhancing model performance by allowing more computational steps during inference (e.g., longer context windows or more decoding steps). This corresponds to increasing $t$ in the above theorem, enabling $M(0 \mid x_{1:t})$ to converge more quickly to the true probability $\mu(0 \mid x_{1:t})$.

### 4.4 Few-shot Example Selection Techniques

An excellent theoretical framework should not only provide a reasonable explanation of existing phenomena but also possess the capability to predict unknown scenarios. Based on the theoretical theorem proposed in Section 4.3, we innovatively introduce a sample selection method for few-shot learning, which can significantly enhance model performance.

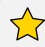 **Takeaway 3**: For few-shot learning, the data points exposing the model's current predictive weaknesses (i.e., lower-confidence predictions) may be more valuable for rapid adaptation than reinforcing already well-learned patterns.

Consider a given computational problem for which there exist multiple computable distributions $\mu_1, \mu_2, \cdots$, that can serve as valid solutions, with their prefix Kolmogorov complexities $K(\mu)$ being approximately equal. Suppose we have collected a large number of sample sequences $x_{1:t}$, where different subsets may originate from different computable distributions. We propose the following sample selection strategy: Prioritize selecting sample sequences $x_{1:t}$

that exhibit a larger difference between $M(0 \mid x_{1:t})$ and $\mu(0 \mid x_{1:t})$. This selection criterion can significantly accelerate the convergence of the predictive model $M(0 \mid x_{1:t})$.

Taking the task of text classification as an example, our method is implemented as follows: among a large number of samples, we prioritize those where the large language model exhibits lower prediction accuracy (more precisely, samples where the model assigns lower probability to the correct next token), rather than those with high prediction accuracy. This selection strategy allows for a more targeted improvement in the model's few-shot learning performance. In summary, this strategy effectively improves the model's performance in few-shot learning scenarios by selectively choosing informative samples.

# 5 EXPERIMENTS

## 5.1 SETUP

We evaluate our few-shot sample selection strategies on three text classification datasets using models from the Qwen2.5 (3B, 7B) (Yang et al., 2024) and Llama (Llama 3.1 8B, Llama 3.2 8B) (Grattafiori et al., 2024) families, and all models used in our experiments are instruction-tuned versions. For each task, specific prompt templates were designed (see Appendix F for examples).

Our methodology for few-shot example selection involves two phases, using a fixed number of 10 few-shot examples. In the first phase (low-confidence selection), we iterate through available samples for each class, identifying those for which the model, given the current prompt, assigns the lowest confidence to the correct label. The confidence is the model's softmax output probability for the ground-truth label token. This process continues until K samples are selected per class, forming the low-confidence set $E_1$. In the second phase (high-confidence selection), serving as a comparative baseline, we similarly select K samples per class with the highest predicted label probabilities to form set $E_2$. These sets $E_1$ and $E_2$ are then used as the few-shot examples, and classification accuracy is evaluated on a held-out test set. Additional details are provided in Appendix D.

---

**Algorithm 1:** Confidence-Based Sample Selection for Few-shot Text Classification

---

**Input:** Dataset $\mathcal{D} = \{(x_i, y_i)\}_{i=1}^n$, Model $\mathcal{M}$, Initial prompt $p$, Number of samples $K$
**Output:** Low-confidence sample subset $\mathcal{D}'$ for few-shot learning
Initialize $\mathcal{D}' \leftarrow \emptyset$ ;                 // Empty set to store selected samples
$p_{\text{current}} \leftarrow p$ ;             // Initialize current prompt with base prompt
**for** $t \leftarrow 1$ **to** $K$ **do**
    min_conf $\leftarrow 1.0$ ;               // Initialize minimum confidence score
    $x_{\text{selected}} \leftarrow$ null ;                 // Sample with minimum confidence
    **for** $(x, y) \in \mathcal{D} \setminus \mathcal{D}'$ **do**
        $p_{\text{temp}} \leftarrow p_{\text{current}} \oplus x$ ;             // Concatenate prompt with sample
        conf $\leftarrow \mathcal{M}(y|p_{\text{temp}})$ ;           // Get probability of correct label
        **if** $conf < min\_conf$ **then**
            min_conf $\leftarrow$ conf;
            $x_{\text{selected}} \leftarrow (x, y)$;
    $\mathcal{D}' \leftarrow \mathcal{D}' \cup \{x_{\text{selected}}\}$ ;             // Add selected sample to output set
    $p_{\text{current}} \leftarrow p_{\text{current}} \oplus x_{\text{selected}}$ ;               // Update prompt with new sample
**return** $\mathcal{D}'$

---

## 5.2 DATASETS

We evaluate our approach on three benchmark text classification datasets. The first is a binary spam classification dataset (Almeida and Hidalgo, 2011). The second is a 6-class emotion recognition dataset (Saravia et al., 2018), and the third is a 4-class news article classification dataset (Zhang et al., 2015). For the datasets provided by Saravia et al. (2018) and Zhang et al. (2015), which already contain predefined training and test splits, we randomly sampled a subset from the original training set as our selection pool while using the official test set for evaluation. Regarding the Almeida and

Hidalgo (2011) dataset that lacks predefined splits, we first partitioned the data into training and test sets before applying the same sampling strategy.

## 5.3 RESULTS AND ANALYSIS

| Model Family | Version | Size | Confidence | SMS | EMOTION | AG NEWS |
|---|---|---|---|---|---|---|
| | | | | Acc. (%) / Mean@10 | | |
| QWEN | 2.5 | 3B | High ↑ | 76.62 | 55.21 | 71.38 |
| | | | Low ↓ | **90.07** | **56.03** | **74.67** |
| | | 7B | High ↑ | 92.73 | 57.58 | 77.09 |
| | | | Low ↓ | **94.60** | **57.68** | **80.35** |
| LLAMA | 3.2 | 3B | High ↑ | 64.94 | 36.40 | 45.98 |
| | | | Low ↓ | **73.22** | **41.86** | **47.34** |
| | 3.1 | 8B | High ↑ | 85.22 | 52.98 | 74.45 |
| | | | Low ↓ | **85.56** | **53.22** | **76.92** |

Table 1: Comparative performance of few-shot example selection strategies on text classification benchmarks. The table displays accuracy (%) for Qwen and Llama model variants on SMS, EMOTION, and AG NEWS datasets when using high-confidence versus low-confidence example selection. Results indicate that selecting low-confidence examples (Low ↓) consistently yields higher accuracy across models and datasets compared to high-confidence selection (High ↑).

The results presented in Table 1 demonstrate that our theoretically-grounded strategy of selecting low-confidence samples for few-shot learning consistently yields significant accuracy improvements across all tested models and datasets. This empirical validation aligns with our hypothesis that exposing models to instances where their current predictive understanding is weakest (lower confidence) fosters more rapid and effective adaptation, a principle echoing the error-correction mechanisms inherent in Solomonoff induction. Notably, while the low-confidence strategy remains superior, the *magnitude* of the performance gain appears to moderate with increasing model scale. This observation might suggest that larger models, possessing greater intrinsic capacity and having learned more robust priors during pre-training, may already have a better initial grasp of the task distribution. Consequently, while they still benefit from the targeted information provided by low-confidence examples, their baseline performance with high-confidence (or even randomly selected) examples is already higher, leading to a less pronounced, though still present, advantage for the low-confidence approach.

## 6 CONCLUSION

This paper establishes a formal link between large language models and Solomonoff's theory of universal induction. We prove that LLM training is a monotonic approach to the Solomonoff prior and, under the assumption $M(x_{1:t}) \approx \overline{M}(x_{1:t})$, that LLM inference approximates Solomonoff induction. This AIT-grounded view unifies key LLM phenomena—in-context learning, few-shot adaptation, and scaling laws—as consequences of approximate universal induction. Our theoretical insights motivate a principled few-shot selection rule: prioritize samples that expose the model's predictive weaknesses (lower-confidence predictions) to accelerate adaptation. Experiments on SMS spam, emotion, and news classification confirm that this strategy significantly outperforms high-confidence selection, with the largest gains for smaller models. Together, these results offer a simple, practical method for improving LLM performance while grounding it in AIT. By bridging empirical LLM success with foundational AIT principles, this work provides both a deeper understanding of these models and actionable strategies for their improvement. We contend that viewing LLMs as a computable surrogate for Solomonoff induction encourages more principled advances in model design and application.

ETHICS STATEMENT

This work is primarily theoretical and is evaluated only on standard, publicly available text-classification datasets using their official splits. We do not collect, annotate, or redistribute human-subject data; to the best of our knowledge these datasets contain no personally identifiable information, and our use complies with their licenses. The proposed method does not target or infer sensitive attributes and does not introduce safety risks beyond common LLM usage; we neither elicit nor disseminate harmful content and discourage any misuse. No new data were created, no IRB approval was required, and there are no conflicts of interest or undisclosed sponsorship. We affirm that this submission adheres to the ICLR Code of Ethics.

REPRODUCIBILITY STATEMENT

We provide formal definitions, dataset sources and preprocessing steps, and training/evaluation protocols in the main text, with full hyperparameters, prompt templates, random seeds, and environment versions in the appendix. An anonymized repository with code, configuration files, and one-click scripts to reproduce all reported tables and figures from a clean checkout is available at: https://anonymous.4open.science/r/LLM-as-Solomonoff-Induction-6580. Deteministic settings and data-conversion utilities are included to facilitate cross-checks and extension by other researchers.

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

# APPENDIX

## A  LIMITATIONS

While our work establishes a novel theoretical connection between LLMs and Algorithmic Information Theory, certain limitations should be acknowledged. Firstly, the proposed theoretical interpretation of LLM training/inference through the lens of the Solomonoff prior/induction is heuristic and qualitative. Solomonoff's framework is uncomputable, and our results demonstrate how LLMs offer a *computable monotonic approach*, which, while powerful, inherently diverges from the theoretical ideal due to practical constraints such as finite model capacity and optimization heuristics. Secondly, our experimental validation of the few-shot example selection strategy, while promising, was conducted on specific text classification tasks and a subset of LLM architectures. Further research is needed to ascertain the generalizability of these findings across a broader range of tasks, modalities, and model scales. Finally, while our theory provides a unifying lens for phenomena like scaling laws and in-context learning, the precise quantification of factors like the Kolmogorov complexity of target distributions ($K(\mu)$) in real-world LLM scenarios remains a complex endeavor, making direct measurement challenging.

## B    IMPACT STATEMENT

This research advances theoretical understanding of LLMs, guiding more principled, efficient development. The AIT connection can inform interpretability, generalization, and data-efficient learning. Our few-shot selection strategy improves LLM performance, especially for smaller models, enhancing accessibility and reducing computational costs. We foresee no direct negative societal impacts from this theoretical work and selection method, as it offers an analytical framework and efficiency gains, not new high-risk capabilities. While any AI advancement could theoretically be misused, we stress that responsible, ethical LLM development is paramount. Our research aims to contribute positively to AI's scientific understanding and responsible progress.

## C    NOTATION AND SYMBOLS

This section provides a summary of the key mathematical notations and symbols used throughout the paper.

- $T$: A specific Turing machine.
- $U$: A universal Turing machine (UTM).
- $p$: A program for a Turing machine.
- $w$: Input for a program $p$ on a TM.
- $U(p, w)$: Output of UTM $U$ with program $p$ and input $w$.
- $U(p)$: Output of UTM $U$ given program $p$.
- $x_{1:t}$: An input sequence of tokens (context) of length $t$.
- $x_i$: The $i$-th token in a sequence.
- $x^*$: A sequence starting with prefix $x$. In LLM generation (Def 1), $x^* = x \circ r$, the full sequence generated from prompt $x$ and model output $r$.
- $\mathcal{V}$: Vocabulary of tokens.
- $P(A|B)$: Conditional probability of $A$ given $B$.
- $P_\theta(x_{t+1}|x_{1:t})$: Conditional next-token probability of an LLM with parameters $\theta$.
- $\ell(p)$: Length of program $p$ in bits.
- $K_U(x)$: Prefix Kolmogorov complexity of string $x$ with respect to prefix UTM $U$.
- $K(x)$: Prefix Kolmogorov complexity of $x$ (UTM $U$ implied).
- $M(x)$: The Solomonoff prior probability of string $x$.
- $M(x_{t+1}|x_{1:t})$: Solomonoff induction predictive probability for the next token $x_{t+1}$ given $x_{1:t}$.
- $g : \mathcal{X} \times \mathcal{S} \to \mathcal{X}^*$: Language Model Generation Function (Definition 1).
- $\mathcal{X}$: Set of input prompts for an LLM.
- $\mathcal{S}$: Set of random seeds for an LLM.
- $\mathcal{R}$: Set of possible model outputs (continuations) from an LLM.
- $s$: A random seed.
- $r$: Output sequence generated by an LLM.
- $\circ$: String concatenation.
- $f(x, s)$: A 4-tuple program $(m_{(2)}, n(x)_{(2)}, s_{(2)}, e(x)_{(2)})$ constructed for an LLM.
- $m_{(2)}$: Binary representation of the core LLM model component.
- $e(x)_{(2)}$: Binary encoding of string $x$ using the LLM and arithmetic coding.
- $n(x)_{(2)}$: Binary representation of the number of decoding iterations for $e(x)_{(2)}$.
- $s_{(2)}$: Binary representation of the random seed $s$.
- $\overline{f}(x, s)$: Prefix-coded version of the program $f(x, s)$, i.e., $(\overline{n}(x)_{(2)}, \overline{s}_{(2)}, \overline{e}(x)_{(2)})$.

- $\overline{n}(x)_{(2)}, \overline{s}_{(2)}, \overline{e}(x)_{(2)}$: Elias gamma coded versions of $n(x)_{(2)}, s_{(2)}, e(x)_{(2)}$ respectively.

- $\ell(\overline{f}(x,s))$: Length of the program $\overline{f}(x,s)$.

- $\overline{M}(x)$: Approximate Solomonoff prior defined as $\sum_{s=1}^{\infty} 2^{-\ell(\overline{f}(x,s))}$.

- $\overline{F}$: The set of all prefix-encoded programs $\overline{f}(x,s)$.

- $U_F$: A prefix UTM defined based on $\overline{F}$.

- $\ell(e(x)_{(2)})$: Length of the binary encoding $e(x)_{(2)}$.

- $\theta$: Parameters of a trained LLM.

- $\theta'$: Parameters of a LLM without trianing.

- $\log_2$: Logarithm to the base 2.

- $\pi$: The mathematical constant pi (approx 3.14159).

- $\mu$: A computable target probability distribution.

- $K(\mu)$: Prefix Kolmogorov complexity of the distribution $\mu$.

- $\mathbb{B}^t$: The set of all binary strings of length $t$.

- $c$: A generic constant.

- $\mathcal{D}$: A dataset, typically a set of pairs $\{(x_i, y_i)\}$.

- $(x, y)$: A data sample (input $x$, label $y$).

- $\mathcal{M}$: A large language model.

- $p$ (in Algorithm 1): Initial prompt.

- $K$ (in Algorithm 1): Number of few-shot samples to select.

- $\mathcal{D}'$: Subset of data selected for few-shot learning.

- $p_{\text{current}}$: Current prompt being constructed.

- conf: Confidence score (model's probability for the correct label).

- $\oplus$: Symbol used in Algorithm 1 to denote prompt concatenation.

- $E_1$: Set of low-confidence few-shot examples.

- $E_2$: Set of high-confidence few-shot examples.

# D   DETAILED EXPERIMENTAL CONFIGURATION

All inference tasks for the experiments were conducted on a system equipped with 4 NVIDIA A100 GPUs. The cumulative computation time for running all experiments, encompassing different models, datasets, and both few-shot selection strategies (low-confidence and high-confidence), amounted to approximately 1.5 days.

A critical parameter for our experiments was the decoding temperature, which was uniformly set to 0 for all large language model inference steps. This choice is pivotal for ensuring deterministic outputs. By setting the temperature to 0, we effectively select the most probable token at each step of the generation process, thereby eliminating randomness typically introduced by temperature-based sampling. This determinism is methodologically important for several reasons:

1. **Reproducibility:** It ensures that results are perfectly reproducible given the same model and input.

2. **Alignment with Theoretical Framework:** As discussed in Section 3.1, our theoretical framework views LLMs as specific Turing machines, which are inherently deterministic. Setting temperature to 0 makes the practical LLM behavior more closely approximate this deterministic ideal. The model's output becomes a direct function of its learned parameters and the input sequence, without the confound of stochastic sampling.

3. **Fair Comparison:** It provides a stable baseline for comparing the efficacy of the few-shot selection strategies ($E_1$ vs. $E_2$). Any observed performance differences can be more confidently attributed to the selection strategy itself, rather than variations due to sampling.

This approach allows for a more rigorous evaluation of how different few-shot examples influence the model's underlying predictive tendencies, in line with our goal of understanding LLMs as systems approximating Solomonoff induction, which is itself a deterministic (though uncomputable) predictive framework.

## E  Two lemmas about Turing machines

**Lemma 4** *Let $M$ be a universal Turing machine, and let $F$ be a prefix-free set of programs. Then there exists a universal prefix Turing machine $U_F$ such that every program $p \in F$ is valid on $U_F$ and satisfies:*

$$U_F(p) = M(p)$$

*Proof.* Since $F$ is prefix-free, there exists a finite string $s$ that is not a prefix of any program in $F$. Such a string $s$ exists because $F$ is prefix-free and hence satisfies the Kraft inequality.

We define $U_F$ like this. For any input $p$, If $p \in F$, simulate $M(p)$. Else, check if $p$ starts with $s$. If yes, let $p = s \cdot q$. Simulate $U(q)$, where $U$ is a standard universal prefix Turing machine. If no, $U_F$ diverges (halts with no output).

Hence, $U_F$ is a universal prefix Turing machine that satisfies $U_F(p) = M(p)$ for all $p \in F$.

□

**Lemma 5** *Let $U$ be a universal Turing machine and $S$ be an arbitrary Turing machine. There exists a universal Turing machine $U'$ that satisfies:*

1. *$U'$ embeds $S$ within its structure while maintaining universality*

2. *$U'$ can directly accept inputs intended for $S$*

*Proof.* We can construct $U'$ as follows:

1. $U'$ first examines its input to determine whether it's intended for direct execution by $S$ or for universal simulation.

2. $U'$ reserves a special prefix symbol or sequence (let's call it $s$) to indicate that the remaining input should be directly processed by $S$.

3. $U'$ implements the following algorithm:

   (a) If the input begins with prefix $s$, strip the prefix and run $S$ directly on the remaining input.

   (b) Otherwise, run $U$ on the input as normal.

□

## F  Prompts Used in Experiments

System prompt for SMS:

```
Classify the following SMS message as either spam or ham.
Respond with only one word: "spam" or "ham"
(without quotes or any additional text).
Examples:
{examples}
```

System prompt for EMOTION:

```
## Emotion Classification Task
Classify the following text into one of these six basic emotions:
- sadness
- joy
- love
- anger
- fear
- surprise
## Response Guidelines:
- Respond with only one word--the most relevant emotion from the list above.
- Do not include quotes, punctuation, or any additional text.
- Choose the emotion that best represents the overall sentiment of the text.
## Examples:
{examples}
```

System prompt for AG NEWS:

```
## News topic classification Task
Classify the following news articles into one of these four basic topics:
- World
- Sports
- Business
- Sci/Tech
## Response Guidelines:
- Respond with only one word--the most relevant topic from the list above.
- Do not include quotes, punctuation, or any additional text.
- Choose the topic that best represents the overall sentiment of the text.
## Examples:
{examples}
```

