# OpenReview forum: "Large Language Models as a Computable Surrogate to Solomonoff Induction"
_ICLR.cc/2026/Conference — Submitted to ICLR 2026_

### Official Review · Reviewer_DQVi · 2025-10-16

**Soundness:** 1
**Presentation:** 2
**Contribution:** 1
**Rating:** 2
**Confidence:** 3

**Summary:**

The authors study the behavior of LLMs through Algorithmic Information Theory. They argue that the LLM training process
can be viewed as approximating the Solomonoff prior and, under the assumption that the approximate Solomonoff prior
induced by the model is close to the true one, LLM next-token prediction can be viewed as a form of approximate
Solomonoff induction. Then, based on these observations, they propose an uncertainty-based method to choose examples
in few-shot learning and conduct experiments to show the validity of this method.

**Strengths:**

* Overall, the paper is easy to follow and clear, and the use of those colored boxes does make it easier to locate
  important information.
* The use of Solomonoff complexity in this setting is novel and seems to be interesting.

**Weaknesses:**

The main issue of this paper is that, while the authors claim the paper to be theoretical, the theoretical results
are non-rigorous at best.
* In Theorem 1, the authors claim that the approximate Solomonoff prior induced by the model $\bar{M}(x)$ will approach
  the true Solomonoff prior $M(x)$ as the training loss decreases. However, not only the authors do not define what
  (class of) losses they are considering and provide little justification on this statement. They cite
  [Delétang et al., 2023] and [Wan, 2025] to justify it, but I do not think this is proved anywhere in these two papers.
* The second theoretical part of the paper (approximate Solomonoff induction) relies on the assumption that
  $\bar{M}(x) \approx M(x)$. I do not see why this is reasonable even in some idealized case. The Solomonoff prior $M(x)$
  aggregates all possible programs that generate $x$ as a prefix, while the $\bar{M}(x)$ considers a single model and
  vary only the random seed. Therefore, I do not think $\bar{M}(x)$ can be used to approximate $M(x)$ (unless we interpret the model as a
  universal Turing machine and view the seed as the program to be simulated, which is very different from the usual
  way of interpreting LLMs and this type of behavior cannot be expected from any conventionally-trained LLM).
* In addition, for the empirical part of the paper, uncertainty-based few-shot sample selection is not a new idea and
  has at least appeared in, say, [1].


[1] Shizhe Diao, Pengcheng Wang, Yong Lin, Rui Pan, Xiang Liu, Tong Zhang. Active Prompting with Chain-of-Thought for Large Language Models. 2023.

**Questions:**

See the Weakness section for details.
* Could you further justify point 2 of Theorem 2?
* Could you explain why it is reasonable to expect $\bar{M}(x) \approx M(x)$, at least in some idealized settings?

---

### Official Review · Reviewer_uMSW · 2025-11-02

**Soundness:** 3
**Presentation:** 3
**Contribution:** 3
**Rating:** 2
**Confidence:** 4

**Summary:**

This paper addresses an important question in Transformers — why models struggle to generalize to longer sequences than those seen during training. The authors propose that length generalization (LG) depends on how well the model’s inductive bias aligns with the computational structure of the task. The authors study this question on Polynomial Iteration task, which has clear recursive computational rules. The authors examine how various positional encodings (PEs) — including Absolute (APE), Relative (T5, RoPE), and No Positional Encoding (NoPE) — influence LG. The authors find that Transformers can simulate iterative computation when trained on Polynomial Iteration, but the alignment is fragile under extrapolation. The model's inductive biases - computational (from PEs) and structural (from attention) are misaligned with the inherent computational structure of the task. Interestingly, NoPE sometimes generalizes better than explicit encodings due to implicit positional information in hidden-state statistics and contextual token distributions. The authors aim to reduce the two sources of misalignment: (i) structural bias from softmax attention and (ii) computational bias from PEs. They propose ViPE (Value-side relative position encoding with logic rescaling), which stabilizes the computation and improves LG.

**Strengths:**

1. The observation that LG emerges from alignment between a model’s inductive bias and the computational structure of the task — is both intuitive and interesting.

2. The findings that NoPE models can exhibit partial LG by implicitly encoding positional information through hidden-state mean/variance and contextual token distributions is interesting.

**Weaknesses:**

1. The evolution is performed on synthetic tasks. It would be interesting to see how the approach (particularly ViPE) performs on realistic tasks. Overall, limited evolution that does not cover a larger range of tasks.

2. Limited connections with other existing works on PE (e.g., spectral analysis of PEs).

3. ViPE performance is reported only on polynomial iteration task. Since the authors aim to reduce the two sources of misalignment: (i) structural bias from softmax attention and (ii) computational bias from PEs and propose ViPE (Value-side relative position encoding with logic rescaling), it would be good to evaluate this comprehensively.

4. The propositions presented in the paper rely on uniform attention, which is used as a simplifying assumption but it is not clear to what extent does limit the realism.

**Questions:**

How would the approach perform of other more realistic tasks, e.g., NLP tasks?

How does the approach connects to other PEs-based modeling?

How sensitive are the results to the assumption of uniform attention?

---

### Official Review · Reviewer_1Ner · 2025-11-03

**Soundness:** 1
**Presentation:** 2
**Contribution:** 1
**Rating:** 2
**Confidence:** 3

**Summary:**

The authors analyze the well-known properties and capabilities of autoregressive LLMs through the lens of Solomonoff priors and induction. In particular, they claim that LLM training can be framed as learning a computable approximation to the Solomonoff prior over language sequences (i.e., log-loss minimization in LLM training is implicitly a search over minimum description-length programs that generated the data), and that this result implies that LLM inference can be interpreted as approximate Solomonoff induction. Based on these connections, they analyze the in-context learning behavior of LLMs. They propose a few-shot example selection approach based on their framework and empirically validate its effectiveness on a small number of text-based prediction tasks.

**Strengths:**

- The paper proposes a theoretical framework for understanding in-context learning for autoregressive LLMs.
- Based on the theoretical connections claimed in the paper, the authors propose and empirically validate a method for few-shot example selection, which led to improved performance on a few text-based prediction tasks.
- The paper provides succinct descriptions of relevant background and the overall setup in an accessible manner, and the writing is generally clear and easy to follow.

**Weaknesses:**

- While the discussed connections to Solomonoff induction are conceptually appealing, the core finding seems to be that language modeling reduces to compression (last paragraph of Section 4.1). This has been noted in prior works [1] and therefore potentially limits the novelty of findings. I think the paper could benefit from explicitly articulating their main novel contributions in contrast to existing works in this space.
- An aspect that is central to the analysis seems to be the assumption that $M(x_{1:t}) \approx \bar{M}(x_{1:t})$, but this claim is not theoretically or empirically justified. In particular, Section 4.1 provides a conceptual recipe for how one could construct computable approximations to the Solomonoff prior from LLMs, but it is unclear whether this is indeed a "good" approximation. Subsequent sections hinge on whether this assumption, and I think the theoretical grounding is overall somewhat weak.
- Empirical support of the main findings of the paper are limited. The only experiment included is on their few-shot example selection approach, which only pertains to one specific implication of the proposed framing and limited in scope.
- The notations used throughout the paper could be improved for better clarity.

References:
[1] Language Modeling is Compression (Delétang et al., 2024)

**Questions:**

Please see the Weaknesses section.

---

### Official Review · Reviewer_TAUy · 2025-11-03

**Soundness:** 1
**Presentation:** 1
**Contribution:** 2
**Rating:** 2
**Confidence:** 2

**Summary:**

This paper points out that explaining the powerful emergent capabilities of LLMs through a unified mathematical framework remains a challenge and further proposes a formal theoretical framework to bridge the LLM architectures and algorithmic information theory. The paper proves two essential insights regarding LLMs: (1) the training process of LLMs computationally approaches Solomonoff prior, and (2) LLMs' next-token prediction implements a form of surrogate Solomonoff induction.

**Strengths:**

- This paper focuses on theoretical understanding of LLMs' mechanisms, which is an essential problem.
- The paper bridges the LLM architectures and algorithmic information theory, which provides a new perspective of understanding LLMs.

**Weaknesses:**

- The writing of the paper is not clear and should be further polished. For example, since $M$ is not defined before, it is confusing to see it in the Abstract section.
- The paper seems to overclaim its contribution. The connection between the Turing Machine and LLMs is not clear. The authors just view the LLMs as Turing Machines and lack the corresponding explanations or demonstrations.
- The proofs are not well managed and arranged.

**Questions:**

- The definition in Section 3 formulates the mechanism of LLMs in a text completion mode. Is this definition established for all cases in this paper?
- What is the working mechanism of a prefix UTM? What is the connection between the prefix UTM and LLMs? It would be better to provide clear explanations in the paper.

---

### Official Review · Reviewer_4cuM · 2025-11-04

**Soundness:** 3
**Presentation:** 3
**Contribution:** 2
**Rating:** 4
**Confidence:** 2

**Summary:**

This paper introduces a novel theoretical framework arguing that Large Language Models (LLMs) can be formally understood as computable surrogates for the uncomputable Solomonoff induction. The paper's primary contribution is establishing this link via two claims: first, that LLM training is a computable approach to the Solomonoff prior by reframing loss minimization as program length optimization, and second, that LLM inference implements surrogate induction. This AIT-based theory motivates a practical and counter-intuitive few-shot selection strategy: prioritizing samples where the model has the *lowest* predictive confidence. This low-confidence method is empirically validated on text classification benchmarks, demonstrating significant performance gains over a high-confidence baseline.

**Strengths:**

Reasons to Accept

- High Novelty: The paper establishes the first formal, constructive link between modern LLM architectures and the foundational principles of Algorithmic Information Theory (AIT).
- Explanatory Power: The AIT framework provides a unified lens for explaining disparate emergent phenomena like in-context learning, few-shot adaptation, and scaling laws.
- Practical Contribution: The deep theory leads directly to a practical, counter-intuitive, and effective few-shot selection algorithm (low-confidence sampling).
- Strong Empirical Validation: The low-confidence selection method consistently and significantly outperforms the high-confidence baseline across all tested models (Qwen, Llama) and datasets (SMS, EMOTION, AG NEWS).

**Weaknesses:**

Reasons to Reject


- Theory-Experiment Mismatch: There's a disconnect between the universal scope of the AIT theory and the narrow scope of the experiments (three simple text classification tasks).
- Simpler Alternative Explanations: The success of the low-confidence method can be explained by simpler, well-known concepts like "hard negative mining," which the authors fail to discuss or rule out.
- The paper's most critical limitation (the $M \approx \overline{M}$ assumption) is relegated to Appendix A 1 rather than being addressed upfront in the main paper.

**Questions:**

1.  In the derivation of Theorem 3 (Section 4.2), what is the origin and justification for the $2t$ term in the approximation for $|e(x_{1:t})_{(2)}|$?
2.  How do you disentangle your AIT-based explanation for the success of low-confidence sampling from simpler, established concepts like hard negative mining? A comparison against a *random selection* baseline seems essential and is currently missing.
3.  Have you tested your few-shot selection method on more complex, "algorithmic" reasoning tasks (e.g., math or code generation) where a "universal induction" framework would be more meaningfully tested?

---

### Meta-Review · Area_Chair_ZFNA · 2025-12-12

**Summary:**

This paper presents a unified theoretical framework linking LLMs to algorithmic information theory, establishing formal connections to Solomonoff induction. It shows that LLM training approximates Solomonoff priors and that next-token prediction acts as surrogate Solomonoff induction, which offers a coherent explanation for in-context learning, few-shot learning, and scaling laws. The theory further motivates a low-confidence–based few-shot selection strategy, which empirically improves performance.

This paper received reviewing comments from five reviewers. Some reviewers affirmed the importance and potential of the research question. However, several concerns remained.
- The theoretical results and corresponding discussions should be more rigorous. For the current form, the core assumption is not discussed carefully, e.g., the reasonableness of the assumptions. This has led readers to question the validity of the theoretical results. The relationship between the theoretical results derived in this work and those in previous works is not clear enough, which, to some extent, weakens the technical contribution.
- Experiments are not convincing. Although it seems the author intends to devote more space to theory, the empirical verification is a bit weak. The included tasks and modern models are limited. More empirical results and ablation studies should be offered to enhance the connection between theory and practice.
- The presentation could be polished. Reviewers have raised a series of questions about unclear descriptions of this work, which should be addressed.

Given these issues, the AC recommends rejection. The authors are encouraged to address the mentioned concerns to enhance the clarity and overall impact of the work in future submissions.

**Reviewer Concerns:**

There is no response provided by the authors. Therefore, no concern raised by the reviewers could be addressed. The concerns are still outstanding.

**Reviewer Scores:**

As there is no response provided, reviewers will not participate in discussions with the authors. Their scores will most likely remain unchanged in the end.

---

### Decision · Program_Chairs · 2026-01-26

Reject